# Giant Squid (*Dosidicus gigas*) Meal in Chicken Diets to Enrich Meat with n-3 Fatty Acids

**DOI:** 10.3390/ani12172210

**Published:** 2022-08-27

**Authors:** Jesús Morales-Barrera, María Carranco-Jáuregui, Guillermo Téllez-Isaías, Ana Sandoval-Mejía, Mariano González-Alcorta, Silvia Carrillo-Domínguez

**Affiliations:** 1Departamento de Producción Agrícola y Animal, Universidad Autónoma Metropolitana, Unidad Xochimilco, Calzada del Hueso No. 1100, Col. Villa Quietud, Ciudad de México 04960, Mexico; 2Departamento de Nutrición Animal, Instituto Nacional de Ciencias Médicas y Nutrición Salvador Zubirán, Vasco de Quiroga No.15, Col. Belisario Domínguez Sección XVI, Ciudad de México 14080, Mexico; 3Department of Poultry Science, University of Arkansas, Fayetteville, AR 72704, USA; 4Departamento de Zootecnia, Universidad Autónoma Chapingo, Carretera México-Texcoco, Km 38.5, Texcoco 56230, Mexico

**Keywords:** giant squid meal, production parameters, n-3 fatty acids, chicken meat, sensory evaluation

## Abstract

**Simple Summary:**

Squid is a product with a high protein content and fatty acids such as eicosapentaenoic (EPA), docosapentaenoic (DPA), and docosahexaenoic (DHA) acids. The main marketed parts are the mantle in the form of a fresh-frozen fillet, the head with tentacles (“dancers”), and fins. However, there is another market that can provide added value to the squid fishery, and it focuses on using the viscera, other parts of the squid, or the whole squid that do not meet quality standards for human consumption to produce meal for poultry feeding. The objective of this study was to include giant squid meal (1.67%, 3.34%, and 5.01%) in broiler rations to enrich its meat with n-3 fatty acids. The results showed that squid meal increased the content of EPA, DPA, and DHA in the poultry meat, mainly in females, and the legs with thighs without negatively affecting the production parameters or the flavor of the meat. The use of giant squid meal in chicken diets could benefit not only the poultry sector, also the fishing industry and health-conscious consumers.

**Abstract:**

The main marketed parts of squid are the mantle, the head with tentacles, and fins. However, when the whole squid does not meet quality standards for human consumption it can be used for broiler feed. The objective of the study was to include giant squid (*Dosidicus gigas*) meal (GSM) in broiler rations to increase the content of the n-3 fatty acids eicosapentaenoic (EPA), docosapentaenoic (DPA), and docosahexaenoic (DHA) in chicken meat. Two hundred Ross 380 chickens, half male, half female, and one day old, were randomly distributed in a 4x2x2 factorial arrangement. The factors were the treatment (0%, 1.67%, 3.34%, and 5.01% of GSM in the diet), sex, and content of n-3 in the legs with thighs and the breasts. Each treatment had five repetitions with 10 birds each. There were no differences (*p* > 0.05) in the production parameters for both sexes. The contents of EPA, DPA, and DHA increased in the females and in the legs with thighs (*p <* 0.05) with GSM. Acceptance for the flavor and texture of meat was higher in the treatment with 1.67% GSM than in the other treatments. It is concluded that GSM is an alternative for increasing the amount of n-3 in chicken meat.

## 1. Introduction

The lack of uses for marine products and byproducts, such as squid, represents an economic loss in the regions where this cephalopod is caught. Worldwide, Mexico is among the main countries producing the giant squid (*Dosidicus gigas*), reporting a national production of 2598 tons per year, with the state of Sinaloa producing the most at 1600 tons [1]. The consumption of squid in Mexico is estimated at 0.53 kg/year per capita, which represents 3.8% of the consumption of marine products, well below countries such as South Korea, Japan, and Spain that consume more than 3.5 kg/year per capita. Squid is a product with a high protein content and fatty acid content that is also inexpensive [1].

The mantle in the form of a fresh-frozen fillet, the head with tentacles (“dancers”), and fins are the squid parts that are marketed [2]. In addition to the aforementioned squid product market, there is another market that can provide added value to the squid fishery, and it focuses on using the viscera, other parts of the squid, or the whole squid that do not meet quality standards for human consumption to produce meal for animal consumption. Squid is a rich source of protein and eicosapentaenoic (EPA), docosapentaenoic (DPA), and docosahexaenoic (DHA) fatty acids that are not synthesized by the human body and are considered essential [3]. Currently, consumers concerned about improving their health are looking for foods that contain beneficial bioactive compounds. Among these compounds, the fatty acids DHA and EPA are important for the development of the brain and retina during the last trimester of gestation and the first years of life. These fatty acids also have hypocholesterolemic, anti-inflammatory, antiplatelet, and immunomodulatory properties, among others, which are useful in the prevention and treatment of chronic degenerative diseases [3].

On the other hand, Mexico ranks 12th worldwide as a consumer of chicken meat, consuming 33.12 kg/year per capita, and ranks 6th in global chicken production (3.6 billion tons/year) [4]. Thus, the use of giant squid meal in chicken diets could benefit not only the poultry sector but also the fishing industry.

It is necessary to promote the advantages of this product since it can be incorporated into chicken feed and provide added value to their meat in such a way that it benefits the consumer. However, a few studies have been carried out on poultry farming using squid meal, such as Carranco-Jáuregui et al. [5] who included 5% and 10% squid meal in hen diets and noted that it was not possible to detect the taste or smell of fish in the eggs. On the other hand, Larmond et al. [6] reported that when squid meal was included at 5%, 10%, and 15% in chicken diets, they detected the taste and smell of fish in the meat with 10 and 15% of the meal. Based on the background described, the objective of this research was to evaluate whether the inclusion of giant squid (*D. gigas*) meal in broiler rations enriched its meat with n-3 fatty acids.

## 2. Materials and Methods

### 2.1. Obtaining the Meal

Giant squid meal (GSM) was obtained from whole specimens (mantle, viscera, feather, and tentacles) captured off the coast of Santa Rosalía, Baja California Sur, which is located at 27°20′ N and 112°16′ W at 40 masl and has an average temperature of 22.6 °C. The process for drying and grinding as well as its chemical composition is described by Calvo et al. [3].

### 2.2. Experimental Design

The study was conducted according to the guidelines of the Declaration of Helsinki and approved by Internal Committee for Care and Use of Experimental Animals (CICUAE) of the National Autonomous University of Mexico (UNAM). Ethical approval code: CICUAEFESC C20_06 [7].

A total of 200 broilers (100 male and 100 female) of the Ross 380 lineage at one day of age were used and distributed in four treatments with five repetitions of 10 birds each, 5 males and 5 females. The birds were housed in floor cages of 1.5 × 2 m, with a density of 10 birds per cage, each with a feeder and drinker. The temperature conditions were 25 °C on average.

Four diets were formulated (Table 1) based on sorghum + soybean meal, according to the nutritional needs of the management manual for broilers of the Ross lineage The specifications are a guide and were adjusted to local conditions and markets. Due to the presence of ascites syndrome in the study area, the manual specifications were adjusted at 2250 masl [8].

The control diet was supplemented with GSM at 1.67%, 3.34%, and 5.01%, partially replacing soy meal, using the protein and amino acid values reported by Calvo et al. [3], leaving the isocaloric and isoproteic diets for starter (1–7 days), growth (8–14 days), and finisher (15–49 days). Water and feed were provided ad libitum. During the experimentation period, the production parameters (feed consumption, weight gain, and feed conversion) were measured.

At the end of the experiment (49 days), the 200 birds were sacrificed according to the methods described in the Official Mexican Standard, Humane slaughter of domestic and wild animals (NOM-062-ZOO-1999) [9].

Subsequently, all the carcasses were gutted and deboned, randomly separating 10 legs with thighs and 10 breasts for chemical analysis and the same amount for the sensory evaluation test.

### 2.3. Analysis of Total Lipids and Fatty Acid Profile of Chicken Legs with Thighs and Breasts

Each leg with a thigh and each breast were ground using an Oster food processor for the analysis of total lipids and fatty acids following the methods described by the Association of Official Agricultural Chemists (AOAC) [10].

### 2.4. Sensory Evaluation

The chicken pieces in each treatment were cooked separately in water for 30 min (without adding salt and without any other seasoning). They were removed and allowed to cool and then the meat was shredded.

In the sensory evaluation, 30 untrained judges (both sexes), habitual chicken consumers, participated. A preference test (like or dislike) was used to evaluate the flavor and texture of the meat. Each participant was presented with a plate of small samples of a leg with a thigh from each treatment; the same procedure was used with the breast. In both cases, the judges were given water and white bread to consume between each sample to eliminate residual flavors [11].

### 2.5. Statistical Analysis

The data of the production variables, total lipids, and fatty acids of the meat were analyzed using an ANOVA for a completely random design with a factorial arrangement of 4 × 2 × 2 for the following factors: treatment, sex, and meat. For the comparison of means, the Tukey test was used (*p* < 0.05), and the statistical package SAS [12] was used. The analysis of the data obtained during the sensory evaluation was performed with the nonparametric Friedman test [11]. The statistical model is as follows:
$$Yijk=µ+Ai+Bj+\left(AB\right)ij+Ck+\left(AC\right)ik+\left(ABC\right)ijk+Eijkl\;\{\begin{array}{c} i=1\ldots a\\ j=1\ldots a\\ k=1\ldots c\\ l=1\ldots r \end{array}$$
where:
*Yijk*: response variable in repetition *k*, level *j* of *B*, level *i* of *A*;*µ*= overall average;*Ai* = effect of factor *A* at level *i*;*Bj* = effect of factor *B* at level *j*;*Ck* = effect of factor *C* at level *k*;*(AB)ij* = effect of the *AB* interaction at level *i*, *j*;*(AC)ik* = effect of *AC* interaction at level *i*, *k*;*(ABC)ijk* = effect of *ABC* interaction at level *i*, *j*, *k*;*Eijkl* = random error.


## 3. Results

### 3.1. Production Parameters

Table 2 shows that for feed consumption, weight gain, and feed conversion, there were no significant differences (*p* > 0.05) between treatments; however, between sexes (*p* < 0.05), feed consumption and weight gain were greater in males.

Table 3 shows that the total lipid content in chicken meat was not affected by the inclusion of GSM in the diets (*p* > 0.05). However, the contents of saturated fatty acids (SFAs), monounsaturated fatty acids (MUFAs), and polyunsaturated fatty acids (PUFAs) were higher in the treatment with 1.67% GSM (*p* < 0.05) than in the other treatments. The same behavior was observed for the total content of n-6 and n-3 (Table 4).

The total lipid (TL) content in meat was not affected (*p* > 0.05) by the sex (Table 3). However, the contents of SFAs, MUFAs, and PUFAs were present in higher concentrations in the meat of females than in that of males (*p* < 0.05), and the same occurred with the total n-6 and n-3 (Table 4).

Higher contents of TL, SFAs, MUFAs, PUFAs, and total n-6 and n-3 (*p* < 0.05) were observed in the legs with thighs than in the breasts (Table 3 and Table 4).

The n-6/n-3 ratio was lower (*p* < 0.05) in the treatments with 3.34% and 5.01% GSM than in the other treatments. The relationship between sexes was not different (*p* > 0.05). In the meat, the n-6/n-3 ratio was lower in the breasts than in the legs with thighs (*p* < 0.05) (Table 4).

### 3.2. Sensory Evaluation

The results of the sensory evaluation showed a greater preference for the taste and texture of the leg with a thigh than for that of the breast; however, a greater preference was found for the breast in the treatment with 1.67% GSM (Table 5).

## 4. Discussion

### 4.1. Production Parameters

The inclusion of up to 5% of GSM in the diet of chickens, in substitution of soybean meal, did not have adverse effects on the production parameters, and this result differs from that reported by Hulan et al. [13], who with 7.5%, 15%, and 30% redfish meal, at the expense of soybeans, observed lower feed intake and weight gain, and poor feed conversion; but with lower inclusion levels (4%, 8%, and 12%) also found a significant linear decrease in weight gain and food consumption [14].

Other authors [14,15] also reported a decrease in body weight and poor feed efficiency when high amounts of fishmeal were provided to broilers.

Studies in which shrimp, tilapia, crab, or menhaden meals were included in the diet of broilers in concentrations ranging from 10 to 100% found no negative effects on weight gain and feed conversion, but there were negative effects on feed consumption and carcass yield with meal inclusions greater than 30% [16,17,18].

Some authors note that with a greater increase in fish meals or oils, poultry perceive some unpleasant flavor or aroma (fish), which could be a possible cause of low meal or oil consumption. There are few published scientific studies in which fish, crustacean, or mollusk meals have been used in the diet of chickens as a source of n-3 fatty acids. However, it is evident that using concentrations of GSM lower than the concentrations used by the aforementioned authors as a source of n-3 fatty acids avoided adverse effects on broilers [13,14,15,16,17,18,19].

A study carried out with squid viscera oil in laying hens, where it was included at 2% in their feed, reports that it produced a significant increase in EPA (0.24% vs. 0.05%) and DHA (3.25% vs. 1.0%) with respect to the control diet [20]. However, no reports were found on the use of squid meal in broiler feed as a source of fatty acids.

### 4.2. Fatty Acids Present in Meat

#### Effect of Including GSM

In all treatments with GSM, the chicken meat had a higher content of unsaturated fat, and this result is consistent with that reported by Gonzalez-Esquerra and Leeson [21], who noted that, in general, chicken meat presented a higher content of unsaturated fat (33.5% saturated fat, 30.5% monounsaturated, and 32% polyunsaturated). This is an aspect that further favors the consumption of chicken meat, as other meats, such as beef, contain high concentrations of SFAs and low concentrations of polyunsaturated fatty acids (PUFAs) [22,23]. PUFAs are essential since the human body lacks desaturase enzymes, which prevents them from carrying out desaturation reactions; thus, consuming these fats through the diet is essential. The PUFAs arachidonic (C20: 4 n-6 AA) and EPA (C20: 5 n-3 EPA) give rise to eicosanoids (C_20_) that have important physiological and pharmacological activity (prostaglandins, thromboxanes, leukotrienes, and lipoxins) [24,25].

The increase of n-3 fatty acids (mainly EPA and DHA) in the meat in the treatments that included GSM is consistent with that reported by Hulan et al. [14], who by including 7.6%, 15%, and 30% red fishmeal in the diet of broilers, in substitution of soybean meal, observed significant increases in the total fatty acid n-3 (4.6%, 5.5%, and 7.2% of the total fatty acids (TFAs)) compared to that in the control group (2.2% of the TFAs). In a later study [14] carried out by these same authors, they incorporated 4%, 8%, and 12% red fishmeal, and significant increases were also observed in the concentration of FA n-3 in the chicken meat (4.5%, 5.6%, and 6.5% of the TFAs) with respect to that in the control group (3.5% of TFA).

However, it is interesting to note that while Hulan et al. [13,14] found that when n-3 fatty acids increased in meat, n-6 content decreased, in the present study, the total n-6 and n-3 content in meat increased.

### 4.3. Effect of Sex

The effect of sex on the deposition of fatty acids in the meat was notable in the present study. It was interesting to observe that all fatty acids, except DPA (C20: 5 n-3) and DHA (C22: 6 n-3), were present at higher concentrations in females than in males. This result differs in some aspects from that reported by Hulan et al. [14], who observed a lower deposition of SFAs but a higher deposition of FA n-3 in the meat of females.

Rondelli et al. [26] did not observe an effect of sex on the deposition of fatty acids in abdominal fat when they incorporated 3% soybean oil, 3% chicken fat, or 3% bovine fat.

### 4.4. Effect of the Chicken Meat Part

In the present study, the amount of total lipids and fatty acids, except for DPA (C22: 5 n-3) and DHA (C22: 6 n-3), was higher in the leg with a thigh than in the breast. These results agree with those observed by Hulan et al. [13,14], who reported a higher content of SFAs, MUFAs, and PUFAs (C18: 3 n-3) in the leg with a thigh but a lower content of the other PUFAs (C18: 2 n-6, C20: 4 n-6, C20: 5 n-3, C22: 5 n-3, and C22: 6 n-3). In contrast, a higher concentration of these last PUFAs was found in the breast. Gonzalez-Esquerra and Leeson [21] noted that this may be because the breast has a higher content of phospholipids, and it is in this fraction where most PUFAs are concentrated; however, the leg with a thigh contains a greater amount of triglycerides, and it is in this fraction where SFAs, MUFAs, and PUFAs are preferentially deposited (C18: 3 n-3).

### 4.5. n-6/n-3 Ratio

The n6/n3 ratio obtained for the chicken meat when supplementing the diet of birds with GSM was within that reported by Valenzuela et al. [25], who considered that a favorable ratio ranges from 5/1 to 10/1. However, international organizations recommend a 1–4/1 (n-6/n-3) ratio in the diet. However, currently, the high consumption of n-6 fatty acids has led to maintaining a 10–20/1 ratio, a situation that has contributed to a notable increase in chronic degenerative diseases [24].

Therefore, favoring the consumption of chicken meat enriched with n-3 FA is an alternative for increasing the consumption of these bioactive compounds beneficial for human health, particularly EPA and DHA, to maintain the n-6/n-3 balance. The first is a precursor of eicosanoids, compounds with anti-inflammatory properties, antiplatelet agents, etc.; the latter is an essential component of the cell membranes of the nervous system and the retina. Both participate in the modulation of the immune system [24,27]. Reducing the n-6/n-3 ratio has been associated with a favorable increase in the antioxidant activity of the enzymes superoxide dismutase (SOD), glutathione S transferase (GSH-ST), and glutathione peroxidase (GSH-PX) [24,27].

During the first stages of life for birds, n-3 is vitally important since it modulates the cellular and humoral immune response, as well as inflammation, and maintains the integrity of membranes, thus, preventing infection by pathogenic organisms [27,28].

The American Heart Association [29] recommends that patients with cardiovascular problems consume 1 g/d in supplement form and 2–4 g/d for those with hypertriglyceridemia problems (<200–499 mg/dL. TG). For the remaining population, they recommend consuming 2–3 servings of fish per week to meet EPA and DHA needs. The Food and Agriculture Organization (FAO) suggests a daily consumption of 250 mg of EPA + DHA for adults and 150 mg per day for children [30].

### 4.6. Sensory Evaluation

In this study, the greater acceptance of the meat (leg with a thigh and breast) in terms of the flavor and texture occurred for the treatment with 1.67% GSM, indicating that supplementing a chicken diet with low concentrations of GSM is the best option for obtaining the desired consumption of chicken meat enriched with n-3 FA from the GSM.

Unpleasant flavors and odors have been reported in chicken meat when high levels of GSM are used in the diet of birds, so it is recommended that the inclusion is less than 10% [6].

## 5. Conclusions

It is concluded that substituting up to 5% soybean meal with GSM in the diet of broilers is an alternative for increasing the concentration of n-3 fatty acids in the meat without affecting the production parameters. However, the acceptance of the product was higher when lower substitution levels were used (1.67%).

## Figures and Tables

**Table 1 animals-12-02210-t001:** Composition of the ingredients and nutrients in the experimental diets.

	Starter (1–7 days)				Growth (8–14 days)				Finisher (15–49 days)			
Ingredients	0%	1.67%	3.34%	5.01%	0%	1.67%	3.34%	5.01%	0%	1.67%	3.34%	5.01%
Sorghum	56.369	56.369	56.369	56.369	61.029	61.029	61.029	61.029	64.623	64.623	64.623	64.623
Soybean meal	33.797	32.127	30.457	28.787	28.472	26.802	25.132	23.462	25,571	23.901	22.231	20.561
Soybean oil	4.557	4.557	4.557	4.557	5.709	5.709	5.709	5.709	5.760	5.760	5.760	5.760
GSM	0.000	1.670	3.340	5.010	0.000	1.670	3.340	5.010	0.000	1.670	3.340	5.010
Salt	0.400	0.400	0.400	0.400	0.0400	0.400	0.400	0.400	0.400	0.400	0.400	0.400
Calcium carbonate	1.120	1.120	1.120	1.120	0.927	0.927	0.927	0.927	0.912	0.912	0.912	0.912
Orthophosphate	1.833	1.833	1.833	1.833	1.643	1.643	1.643	1.643	1.523	1.523	1.523	1.523
L-lysine	0.406	0.406	0.406	0.406	0.349	0.349	0.349	0.349	0.238	0.238	0.238	0.238
DL-methionine	0.403	0.403	0.403	0.403	0.345	0.345	0.345	0.345	0.254	0.254	0.254	0.254
L-threonine	0.205	0.205	0.205	0.205	0.171	0.171	0.171	0.171	0.099	0.099	0.099	0.099
Choline	0.500	0.500	0.500	0.500	0.500	0.500	0.500	0.500	0.160	0.160	0.160	0.160
Mixture min *	0.100	0.100	0.100	0.100	0.100	0.100	0.100	0.100	0.100	0.100	0.100	0.100
Vit mix **	0.250	0.250	0.250	0.250	0.250	0.250	0.250	0.250	0.250	0.250	0.250	0.250
Coccidiostat	0.050	0.050	0.050	0.050	0.050	0.050	0.050	0.050	0.050	0.050	0.050	0.050
Antioxidant	0.010	0.010	0.010	0.010	0.010	0.010	0.010	0.010	0.010	0.010	0.010	0.010
Pigment ***	0.000	0.000	0.000	0.000	0.045	0.045	0.045	0.045	0.050	0.050	0.050	0.050
Calculated analysis												
ME (Mcal/kg)	3.025	3.025	3.025	3.025	3.15	3.15	3.15	3.15	3.20	3.20	3.20	3.20
Crude protein (%)	20.0	20.0	20.0	20.0	18.0	18.0	18.0	18.0	17.0	17.0	17.0	17.0
Calcium (%)	1.05	1.05	1.05	1.05	0.90	0.90	0.90	0.90	0.85	0.85	0.85	0.85
Phosphorus (%)	0.05	0.05	0.05	0.05	0.45	0.45	0.45	0.45	0.42	0.42	0.42	0.42
Lysine (%)	1.43	1.43	1.43	1.43	1.24	1.24	1.24	1.24	1.06	1.06	1.06	1.06
Methionine (%)	0.69	0.69	0.69	0.69	0.60	0.60	0.60	0.60	0.50	0.50	0.50	0.50
Met + Cys (%)	1.07	1.07	1.07	1.07	0.95	0.95	0.95	0.95	0.83	0.83	0.83	0.83

* mg/kg of diet: zinc, 50; copper, 12; iodine, 0.300; cobalt, 0.200; iron, 110; selenium, 0.100; manganese, 110. ** kg of diet: vitamin A, 12,000 IU; vitamin D3, 25,000 IU; vitamin E, 30 IU; vitamin K3, 2 mg; thiamine, 2.25 mg; riboflavin, 7500 mg; vitamin B6, 3500 mg; vitamin B12, 0.020 mg; niacin, 45 mg; pantothenic acid, 12.5 mg; biotin, 0.125 mg; folic acid, 1500 mg. *** 15 g/g xanthophylls (*Tagetes erecta*). GSM: giant squid meal.

**Table 2 animals-12-02210-t002:** Production parameters of the broiler experimental diets.

Feed Consumption (kg)					Mean ± SEM
GSM	0%	1.67%	3.34%	5.01%	
Male	5.35 ± 0.12 ^ab^	5.55 ± 0.27 ^a^	5.16 ± 0.08 ^ab^	5.14 ± 0.02 ^ab^	5.30 ± 0.08 ^a^
Female	5.04 ± 0.05 ^ab^	4.97 ± 0.13 ^ab^	4.69 ± 0.17 ^b^	4.90 ± 0.15 ^ab^	4.90 ± 0.06 ^b^
Mean ± SEM	5.19 ± 0.08 ^a^	5.26 ± 0.18 ^a^	4.92 ± 0.13 ^a^	5.01 ± 0.08 ^a^	
Weight gain (kg)					
Male	2.91 ± 0.07 ^ab^	3.03 ± 0.18 ^a^	2.89 ± 0.09 ^ab^	2.79 ± 0,02 ^ab^	2.90 ± 0.05 ^a^
Female	2.68 ± 0.06 ^ab^	2.57 ± 0.05 ^b^	2.58 ± 0.07 ^ab^	2.62 ± 0,09 ^ab^	2.61 ± 0.03 ^b^
Mean ± SEM	2.79 ± 0.06 ^a^	2.79 ± 0.13 ^a^	2.73 ± 0.08 ^a^	2.70 ± 0,05 ^a^	
Feed conversion (kg:kg)					
Male	1.83 ± 0.03	1.83 ± 0.03	1.76 ± 0.03	1.83 ± 0.03	1.86 ± 0.01
Female	1.86 ± 0.03	1.93 ± 0.03	1.80 ± 0.05	1.86 ± 0.03	1.86 ± 0.02
Mean ± SEM	1.85 ± 0.02	1.88 ± 0.03	1.78 ± 0.03	1.85 ± 0.02	

GSM = giant squid meal. The mean and standard error of the mean (SEM) are presented. Five repetitions of 10 birds each. ^a,b^ different letters in each row and column indicate significant differences (*p* < 0.05).

**Table 3 animals-12-02210-t003:** Effect of treatment, sex, and meat on the content of total lipids and saturated, monounsaturated, and polyunsaturated fatty acids when incorporating giant squid (*Dosidicus gigas*) meal into the diet of the broilers.

Factor	Total Lipids (g/100 g)	Saturated Fatty Acids (mg/100 g)	Monounsaturated Fatty Acids (mg/100 g)	Polyunsaturated Fatty Acids (mg/100 g)
Treatment				
0%	6.32 ^a^	790.64 ^b^	956.63 ^bc^	1025.77 ^c^
1.67%	6.10 ^ab^	1098.99 ^a^	1296.10 ^a^	1555.06 ^a^
3.34%	5.70 ^b^	783.10 ^b^	1033.30 ^b^	1374.83 ^b^
5.01%	5.84 ^ab^	685.25 ^c^	897.49 ^c^	117.738 ^c^
SEM	0.13	25.99	34.86	40.43
*p*	0.0084	0.0001	0.0001	0.0001
Sex				
Male	5.87 ^a^	743.72 ^b^	873.70 ^b^	1097.46 ^b^
Female	6.11 ^a^	935.27 ^a^	1218.05 ^a^	1454.22 ^a^
SEM	0.09	18.39	24.63	28.59
*p*	0.0720	0.0001	0.0001	0.0001
Meat				
Leg/thigh	8.63 ^a^	1214.78 ^a^	1616.78 ^a^	1957.33 ^a^
Breast	3.35 ^b^	464.22 ^b^	474.98 ^b^	594.36 ^b^
SEM	0.09	18.36	24.65	28.59
*p*	<0.0001	0.0001	0.0001	0.0001
Interactions				
Treatment × sex	0.0007	<0.0001	<0.0001	<0.0001
Treatment × meat	0.1673	<0.0001	<0.0001	<0.0001
Sex × meat	0.9854	<0.0001	<0.0001	<0.0001
Treatment × sex × meat	0.0415	<0.0001	<0.0001	<0.0001

n = 6. ^a,b,c^ different letters in each column indicate significant differences (*p* < 0.05).

**Table 4 animals-12-02210-t004:** Effect of treatment, sex, and leg/thigh breast on the content of n-3 and n-6 fatty acids when incorporating giant squid (*Dosidicus gigas*) meal into the diet of the broilers.

Factor	Linoleic Acid n-6 (mg/100 g)	Alpha-linolenic Acid n-3 (mg/100 g)	Arachidonic Acid n-6 (mg/100 g)	EPA n-3 (mg/100 g)	DPA n-3 (mg/100 g)	DHA n-3 (mg/100 g)	n-6 (mg/100 g)	n-3 (mg/100 g)	n-6/n-3 (mg/100 g)
Treatment									
0%	816.10 ^c^	100.01 ^c^	58.62 ^c^	2.81 ^c^	13.84 ^d^	7.88 ^b^	874.72 ^c^	124.54 ^c^	7/1 ^a^
1.67%	1240.32 ^a^	144.20 ^a^	87.00 ^a^	6.39 ^a^	22.83 ^a^	15.56 ^a^	1327.32 ^a^	188.99 ^a^	7/1 ^a^
3.34%	1103.82 ^b^	127.96 ^b^	70.32 ^b^	4.94 ^b^	20.06 ^b^	16.64 ^a^	1174.15 ^b^	169.61 ^b^	6/1 ^b^
5.01%	926.02 ^c^	106.76 ^c^	56.70 ^c^	4.40 ^b^	16.54 ^c^	15.29 ^a^	982.73 ^c^	142.99 ^c^	6/1 ^b^
SEM	33.30	4.63	2.03	0.28	0.61	0.51	34.61	5.13	0.08
*p*	<0.0001	<0.0001	<0.0001	<0.0001	<0.0001	<0.0001	<0.0001	<0.0001	0.0078
Sex									
Male	872.92 ^b^	100.93 ^b^	63.70 ^b^	4.13 ^b^	17.75 ^a^	13.76 ^a^	936.62 ^b^	136.59 ^b^	6/1 ^a^
Female	1170.22 ^a^	138.53 ^a^	72.62 ^a^	5.13 ^a^	18.88 ^a^	13.92 ^a^	1242.84 ^a^	176.48 ^a^	7/1 ^a^
*p*	<0.0001	<0.0001	<0.0001	0.0005	0.0675	0.7672	<0.0001	<0.0001	0.4684
Meat									
Leg/thigh	1611.39 ^a^	190.92 ^a^	76.52 ^a^	5.94 ^a^	18.87 ^a^	12.99 ^b^	1687.91 ^a^	228.74 ^a^	7/1 ^a^
Breast	431.74 ^b^	48.54 ^b^	59.80 ^b^	3.32 ^b^	17.76 ^a^	14.69 ^a^	491.55 ^b^	84.33 ^b^	5/1 ^b^
*p*	<0.0001	<0.0001	<0.0001	< 0.0001	0.07370	0.0012	<0.0001	<0.0001	<0.0001
Interactions									
Treatment × sex	<0.0001	<0.0001	<0.0001	<0.0001	<0.0001	<0.0001	<0.0001	<0.0001	0.6374
Treatment × meat	<0.0001	<0.0001	0.2029	<0.0001	0.9113	0.3113	<0.0001	<0.0001	0.0084
Sex × meat	<0.0001	<0.0001	<0.0001	0.1036	<0.0001	0.0018	<0.0001	<0.0001	0.1479
Treatment × sex × meat	<0.0001	<0.0001	<0.0001	0.1036	<0.0001	<0.0001	<0.0001	<0.0001	0.8443

n = 6. ^a,b,c,d^ Different letters in each column indicate significant differences (*p* < 0.05). EPA = eicosapentaenoic acid; DPA = docosapentaenoic acid; DHA = docosahexaenoic acid.

**Table 5 animals-12-02210-t005:** Results of the sensory evaluation (flavor and texture) for the breast and leg with a thigh of broilers fed giant squid (*Dosidicus gigas*) meal in their diets.

		0% GSM	1.67% GSM	3.34% GSM	5.01% GSM
Flavor (%)	Breast	68.7 ^b^	81.2 ^a^	68.7 ^b^	62.5 ^c^
	Leg with thigh	62.5 ^b^	68.7 ^a^	43.7 ^c^	31.2 ^d^
Texture (%)	Breast	25.0 ^c^	68.7 ^a^	43.7 ^b^	12.5 ^d^
	Leg with thigh	62.5 ^b^	75.0 ^a^	50.0 ^c^	50.0 ^c^

n = 30. ^a,b,c,d^ Different letters in each row and column indicate significant differences (*p* < 0.05).

## Data Availability

Not applicable.

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
