# Peer review of "Giant Squid (*Dosidicus gigas*) Meal in Chicken Diets to Enrich Meat with n-3 Fatty Acids"

_animals, 2022, doi:10.3390/ani12172210_

Round 1
Reviewer 1 Report
General Comments:
- Overall, a well-written, interesting, and relevant manuscript.
- Key words can be increased.
- The use of “productive” throughout the manuscript would have to be revised
- References are less than required (30)
Specific comments
· Line 14 this sentence may flow better if its: The objective of the study was to include giant squid in broiler rations to increase the content of n-3 fatty acids eicosapentaenoic (EPA), docosapentaenoic (DPA) and docosahexaenoic (DHA) in chicken meat.
- Line 20 and 21: productive parameters or production parameters? I recommend production or live performance parameters
- Line 26: May want to add more key words to at least 5
- Line 36-37: May want to rephrase this sentence “The main marketed squid parts are the mantle in the form of a fresh-frozen fillet, the head with tentacles (“dancers”), and fins”
- Table 38-41: Make into separate sentence “In addition to the aforementioned squid product market, there is another market that can provide added value to the squid fishery, and it focuses on using the viscera, other parts of the squid, or the whole squid that do not meet quality standards for human consumption to produce meal for animal consumption”.
- Line:54-55: Make this into a separate sentence “it is necessary to promote the advantages of this product since it can be incorporated into chicken feed and provide added value to their meat in such a way that it benefits the consumer”.
- Line:81-83 Rephrase the sentence this sentence to avoid confusion “Maybe the soybean meal of the control diet was supplemented with GSM at 1.67%, 3.34% and 5.01% to formulate the diets for the other feed treatments.
- Table 2 should be moved from Materials & Methods to the results section
- Table 2 Title you can replace “productive to production”
- Table 2 include livability/mortality data if you have it.
- Line: 120 Please consider replacing all productive variables used in this paper with production or live performance variables
- Line 122 “pieces of meat” do you mean yield or meat parts? If so, please consider changing it. Throughout the manuscript
- Line 146: What factor? Rephrase this sentence for more clarity
- On discussion are there any other studies looking at giant squid meal? all the studies cited look at other fish, but authors don't discuss the possible difference in fish meal quality.
- Line 168-171/172: Consider combining sentence “Hulan et al. [13], who incorporated 7.5%, 15% and 30% redfish meal, at the expense of soybeans, and observed lower feed intake and weight gain, and poor feed conversion. Subsequently, these same authors [14] conducted another study using lower inclusion levels (4%, 8%, and 12%) but also found a significant linear decrease in weight gain and food consumption”.
- Line198-202/203 Rephrase “ The increase in n-3 fatty acids (mainly EPA and DHA) in the chicken meat in the treatments that included GSM is consistent with that reported by Hulan et al. [14], who by including 7.6%, 15%, and 30% red fishmeal in the diet of broilers, in substitution of soybean meal, observed significant increases in the total fatty acid n-3 (4.6%, 5.5%, and 7.2% of the total fatty acids (TFAs), respectively) compared to that in the control group (2.2% of the TFAs)”
- Line 207-209. Rephrase “However, it is interesting to note that while Hulan et al. [13,14] showed that when increasing the total concentration of TFAs in chicken meat, that of n-6 decreased, in the present study, the total number of n-6 and n-3 increased”.
- Line 218: revise title chicken piece chicken meat part
- Line 232/233 cite “However, international organizations recommend a 1-4/1 ratio in the diet.”.
Manuscripts submitted to Animals should neither be published previously nor be under consideration for publication in another journal. The main article types are as follows:
- Articles: Original research manuscripts. The journal considers all original research manuscripts provided that the work reports scientifically sound experiments and provides substantial new information. Authors should not unnecessarily divide their work into several related manuscripts, although short Communications of preliminary, but significant, results will be considered. The quality and impact of the study will be considered during peer review.
Articles should have a main text of around 3000 words at minimum and should have more than 30 references. Animals has no restrictions on the maximum length of research manuscripts, provided that the text is concise and comprehensive.
For all Western blot figures, densitometry readings/intensity ratio of each band should be included; the whole Western blot showing all bands and molecular weight markers should be included in the Supplementary Materials.
· https://gcc02.safelinks.protection.outlook.com/?url=https%3A%2F%2Fwww.mdpi.com%2Fjournal%2Fanimals%2Finstructions&data=05%7C01%7C%7C9e5c9fb0d5fd46409bbb08da5fc35e36%7Ced5b36e701ee4ebc867ee03cfa0d4697%7C0%7C0%7C637927590101164745%7CUnknown%7CTWFpbGZsb3d8eyJWIjoiMC4wLjAwMDAiLCJQIjoiV2luMzIiLCJBTiI6Ik1haWwiLCJXVCI6Mn0%3D%7C3000%7C%7C%7C&sdata=bK1Mn1vb1CbvxpjCpno45GWXTsQM%2FJkEYRb4zK4Cd%2FY%3D&reserved=0
Author Response
We thank you very much for the time you have spent on reviewing and editing our manuscript.

Reviewer 2 Report
Jesus and colleagues have determined the effects of giant squid meal in chicken diets on the enrichment of n-3 fatty acids in the meat of chickens. This is an interesting topic. The experimental was well designed and the results were well presented. The major finding is that giant squid meal is an alternative for increasing the amount of n-3 in chicken meat. The following changes could improve the quality of the paper.
1. Line 70, please correct “Figure 1, [3].” To “Figure 1 [3].”.
2. Line 75, what is the meaning of “both sexes”, half male half female? Please specify it.
3. Please simply the figure 1. It should be occupy the entire page.
4. Line 98, please correct “(P <0.05)” to “(P < 0.05)”.
5. Please add the replicates for all the tables and figures in the table note and figure legends.
6. Line 135, please write the P value in italic.
7. Please add the space before and after “<”.
8. Lines 140-141, please revise the figure legends.
9. Please reduce the effective number for the data. Such as correct “1374.83” to “1375”.
10. Line 150-151, please do not use only one sentence as a paragraph. lease check the similar issue throughout the paper.
Author Response

(The authors gave the same response as above.)
